# Keratosis Pilaris-like Eruption during Treatment of Chronic Myeloid Leukemia with Tyrosine Kinase Inhibitors: Literature Review and Report of a Case Related to Imatinib

**DOI:** 10.3390/jcm13010032

**Published:** 2023-12-20

**Authors:** Francesca Ambrogio, Melita Anna Poli, Lucia Lospalluti, Teresa Lettini, Nicoletta Cassano, Gino Antonio Vena, Giuseppe Ingravallo, Gerardo Cazzato, Caterina Foti

**Affiliations:** 1Section of Dermatology and Venereology, Department of Precision and Regenerative Medicine and Ionian Area (DiMePRe-J), University of Bari “Aldo Moro”, 70124 Bari, Italy; dottambrogiofrancesca@gmail.com (F.A.); m.poli94@gmail.com (M.A.P.); l.lospalluti@gmail.com (L.L.); caterina.foti@uniba.it (C.F.); 2Section of Molecular Pathology, Department of Precision and Regenerative Medicine and Ionian Area (DiMePRe-J), University of Bari “Aldo Moro”, 70124 Bari, Italy; teresa.lettini@uniba.it (T.L.); giuseppe.ingravallo@uniba.it (G.I.); 3Dermatology and Venereology Private Practice, 76121 Barletta, Italy; nicoletta.cassano@yahoo.com (N.C.); ginovena@gmail.com (G.A.V.)

**Keywords:** adverse cutaneous reaction, BCR-ABL, chronic myeloid leukemia, keratosis pilaris, imatinib, molecular targets, tyrosine kinase inhibitors

## Abstract

The advent of tyrosine kinase inhibitors (TKIs) blocking BCR-ABL activity has revolutionized the therapeutic management of patients with chronic myeloid leukemia (CML). Adverse cutaneous reactions (ACRs) are common nonhematologic adverse events associated with the use of BCR-ABL TKIs. A characteristic pattern of eruption resembling keratosis pilaris (KP) has been described in patients treated with these drugs, especially nilotinib and dasatinib. The pathogenesis of this ACR is still unknown. This type of reaction appears to be uncommon with imatinib. Here, we report the case of an elderly patient with an asymptomatic KP-like eruption, which appeared one month after starting treatment with imatinib for CML. The case presentation is accompanied by a review of similar reactions in patients with CML treated with BCR-ABL inhibitors, attempting to make an excursus on the molecular targets of such drugs and possible mechanisms underlying this ACR.

## 1. Introduction

Chronic myeloid leukemia (CML) is a myeloproliferative disease associated with the presence of Philadelphia chromosomes (Ph) resulting from the mutual translocation t(9;22) (q34;q11) with the subsequent formation of the *BCR-ABL* fusion gene. Ph is a pathological hallmark of CML, as it accounts for the majority of CML cases [1]. The fusion gene product is a constitutively active tyrosine kinase responsible for uncontrolled cell proliferation, inhibition of apoptosis, and oncogenic progression via various downstream signaling pathways, such as phosphatidylinositol-3 kinase (PI3K)/AKT, mitogen-activated protein kinase (MAPK), and Janus kinase (JAK)/signal transducer and activator of transcription (STAT) pathways [2,3].

The advent of tyrosine kinase inhibitors (TKIs) blocking BCR-ABL activity has revolutionized the therapeutic management of patients with CML. A TKI targeting the BCR-ABL kinase is considered the first-line therapeutic approach to CML [4]. Imatinib mesylate was the first TKI approved for the treatment of CML in the chronic phase. The emergence of resistance to imatinib led to the development of second-generation TKIs (nilotinib, dasatinib, and bosutinib) and the third-generation TKI ponatinib. All these drugs compete with ATP for binding to the ABL1 kinase domain [5].

BCR-ABL TKIs have different side effects, targets, potencies, and effectiveness against distinct *BCR-ABL* mutations. The inhibition of BCR-ABL kinase activity is the main mechanism of these drugs in CML treatment, although they display varying potency against the kinase. For instance, nilotinib has a 30-fold higher potency against BCR-ABL1, and dasatinib has more than 300-fold increased potency of kinase inhibition compared with imatinib in vitro [5,6].

Imatinib and nilotinib were initially regarded as high-affinity specific inhibitors of BCR-ABL, but other targets of these TKIs have been identified, including stem cell factor receptor (KIT), discoidin domain receptors (DDR1 and DDR2), colony-stimulating factor receptor (CSF-1R), platelet-derived growth factor receptors (PDGFR-alpha and -beta), and also nonkinase targets, such as the oxidoreductase NQO2 [7,8,9]. In comparison with imatinib, nilotinib has an improved affinity and selectivity for BCR-ABL and is less potent against PDGFR and KIT [7,9]. Imatinib and nilotinib have a relatively narrow kinase selectivity profile compared with the other ATP-competitive TKIs that have a much broader target spectrum [10]. Dasatinib acts as a multikinase inhibitor against numerous targets, including BCR-ABL, SRC family kinases, KIT, PDGFR, DDR1, DDR2, ephrin receptor tyrosine kinases, TEC family kinases, epidermal growth factor receptor (EGFR), ERK-1/2 and STAT-5 [5,11,12,13]. Bosutinib is a multikinase inhibitor with activity against ABL, SRC, TEC and STE family of kinases, CAMK2G, and ErbB3; however, it does not inhibit KIT and PDGFR [14,15]. Ponatinib is known to target not only BCR-ABL but also RET, FLT3, BRAF, MEKK2, KIT, SRC, DDR1, Tie2, PDGFR, fibroblast growth factor receptor (FGFR), and vascular endothelial growth factor receptor (VEGFR) family members [16,17]. The differential target spectra of TKIs can explain differences in their off-target effects and toxicities via the non-selective inhibition of other targets [5].

Adverse cutaneous reactions (ACRs) are common nonhematologic adverse events associated with the use of BCR-ABL inhibitors. The pathomechanism underlying skin toxicity is unknown, although several hypotheses have been proposed, involving, for example, the role of keratinocyte injury, release of interleukin (IL)-31 and IL-33, or the effects of BCR protein imbalance [18]. ACRs to imatinib were found to affect 7–89% of patients in different series, and their incidence and severity appear to be dose-dependent. Superficial edema, maculopapular rash, and hypopigmentation are among the most common imatinib-related ACRs [19,20,21]. According to a meta-analysis published in 2013, the incidence of all-grade rash in patients treated for CML and other malignancies was 34.3% with nilotinib and 23.3% with dasatinib [22]. Fewer pigmentary changes and less edema have been reported with second-generation TKIs as compared with imatinib [21]. However, the results of a recent systematic review indicated that there are no significant differences in the incidence of all-grade or high-grade rash between new-generation BCR-ABL inhibitors and a standard dose of imatinib. Moreover, a subgroup analysis showed that, in comparison with imatinib, the incidence of all grades of rash is higher in the nilotinib, bosutinib, and ponatinib groups [18].

As concerns nilotinib, dasatinib and ponatinib, varied rash morphologies have been described, including perifollicular and folliculocentric eruptions, such as lesions resembling keratosis pilaris (KP) [22,23,24,25,26]. This type of ACR does not appear to be related to imatinib [23,24]. We describe a case of KP-like eruption in an elderly patient with CML treated with imatinib.

## 2. Case Presentation

An 80-year-old male presented to our emergency room in November 2022 because of progressive skin changes that appeared one month after starting treatment with imatinib 300 mg/day for CML. The cutaneous manifestations were initially noticed on the scalp and then gradually spread to the face and extremities and were not improved by the application of moisturizers. The patient did not complain of itch or other symptoms but was very worried about his skin rash, which was perceived as the initial sign of a possible serious ACR to imatinib. He did not take any other medications, with the exception of clopidogrel 75 mg/day, started in 2013 for prevention of atherothrombotic events. Personal and familial history of similar rashes, atopy, or KP was absent.

Objective examination revealed multiple tiny, pinpoint, follicular, hyperkeratotic, white-colored lesions without perifollicular erythema affecting the scalp (Figure 1), the face, especially the nose, and the proximal limbs (predominantly the extensor surface of upper limbs). The trunk, axillae, pubic area, hands, feet, and eyebrows were not affected. Dermoscopy disclosed features consistent with follicular hyperkeratosis (Figure 2). A 5 mm punch biopsy was taken from the scalp lesional skin, and the histological examination showed the presence of hyperkeratosis and moderate dilatation of the hair follicles with a modest lymphomononuclear infiltrate (Figure 3). There were not any additional relevant histopathological findings, such as abnormalities in epidermal layers, vacuolar degeneration, spongiosis, dyskeratotic cells, follicular damage or atrophy, perifollicular fibrosis, eosinophilic inclusions, mucin deposition, or increase in trichohyalin granules. Therefore, a diagnosis of KP-like eruption was made based on clinical and histopathological features that allowed differentiation from other skin conditions.

Imatinib administration was temporarily stopped by the patient’s hematologist, and one month after treatment withdrawal, there was a complete clinical remission with concomitant normalization of the dermatoscopic findings (details of scalp condition are shown in Figure 4 and Figure 5). Secondary hair loss was not observed in the areas previously affected by hyperkeratotic changes. The patient suffered from androgenetic alopecia, and scalp hair density was not modified by the follicular eruption. No pharmacological treatments were prescribed for the cutaneous reaction, and during the imatinib-free interval, the patient continued to apply the emollients already used in the past weeks.

The patient was reassured about the nature of the skin eruption, strengthening the importance of imatinib reintroduction. Retreatment with imatinib at the daily dosage of 200 mg caused the reappearance of the same cutaneous manifestations (not shown) after a few weeks, but they were milder and well controlled by regular applications of emollients different from those previously used. The patient was subsequently lost to follow-up.

## 3. Discussion

Our elderly patient developed a KP-like rash during treatment with imatinib for CML. The drug was initially prescribed by the hematologist at a dosage lower than the standard dose. Probably, this choice was made on the basis of the patient’s age and comorbid status. The ACR could be regarded as mild, although it was a source of great concern for the patient. The rash resolved after imatinib withdrawal and reappeared after restarting imatinib, supporting the iatrogenic nature of the event and the causal relationship between the event and the drug. On rechallenge, imatinib was administered at a daily dose lower than the previous one, and this might have contributed to the milder intensity of the reaction in the second episode, along with a more appropriate emollient therapy. Changes in dosing schedules are sometimes required during the treatment of CML with conventional TKIs, and some toxic effects of such drugs can be managed by dose reduction [6,27].

In our patient, erythema, lichenoid changes, and cutaneous symptoms were absent, and histopathological analysis showed findings suggestive of KP. The diagnosis was made taking clinical and histopathological features into account, and such features were useful for the exclusion of other disorders. Indeed, multiple conditions could be considered in the differential diagnosis, including trichodysplasia spinulosa, follicular lichenoid eruptions, such as that reported in Graham–Little–Piccardi–Lassueur syndrome, lichen spinulosus, follicular mycosis fungoides, perforating disorders, scurvy, phrynoderma, pityriasis rubra pilaris, and follicular hyperkeratotic spicules associated with multiple myeloma [28,29,30,31,32,33,34].

KP is a common benign disorder characterized by spiny keratotic papules, mostly distributed on the extensor surfaces of the proximal extremities, that has historically been associated with atopy [34]. An eruption resembling KP has been characterized as a peculiar ACR to BCR-ABL TKIs, particularly nilotinib and dasatinib [22,24]. It was found that lesions are often symptomatic, and their onset is usually within 2–3 months after drug initiation. Such lesions seem to be dose-dependent and disappear after treatment discontinuation [24,26]. Other manifestations suggestive of a folliculocentric process have been observed with nilotinib and dasatinib [26]. A few cases of KP-like eruption have also been noticed during treatment with ponatinib [24].

We performed a literature search in the PubMed database using the keywords “keratosis pilaris” and “tyrosine kinase inhibitor” or “imatinib”, “nilotinib”, “dasatinib”, “bosutinib”, and “ponatinib”. The search included articles in English published up to August 2023. The references of retrieved manuscripts were also evaluated. Reports of KP or KP-like eruption in patients with CML treated with imatinib or newer generation TKIs were selected.

Table 1 summarizes the most important information contained in such reports [22,23,24,35,36,37,38,39,40,41,42].

Many reactions were observed in patients treated with nilotinib, a few reactions during treatment with dasatinib, whereas fewer cases were reported with ponatinib. Interestingly, case reports of KP-like eruption during treatment with imatinib were not present in the PubMed database. Indeed, some authors declared that imatinib was not known to produce this kind of skin eruption [23,24].

As shown in Table 1, most publications on PubMed consist of the description of single cases, whereas very few articles contain data on small case series [23,24].

In particular, Delgado et al. assessed the prevalence and type of ACRs in 39 CML patients treated with nilotinib (72%) and dasatinib (28%), the majority of whom had received imatinib as first-line therapy [23]. These authors emphasized the propensity of second-generation TKIs to affect follicular structures. In fact, in their patient series, 23% of patients (nine subjects) had extensive KP, 41% had scalp alopecia, and 38.5% had body hair loss, strongly suggesting folliculocentric processes not previously described for imatinib. Cutaneous biopsies were performed in four patients with extensive KP. For all, histopathological examination showed follicular atrophy and perifollicular fibrosis.

Patel et al. described nine patients with CML who developed KP-like lesions on the face, trunk, and/or extremities during treatment with new-generation TKIs (four with nilotinib, one with dasatinib, two with ponatinib, and two with both nilotinib and dasatinib) [24]. Such lesions were accompanied by pruritus in most cases and associated in some patients with scarring or nonscarring alopecia that showed clinical and histopathological features reminiscent of lichen planopilaris. The most striking histopathological findings were intrafollicular keratotic plugs and perifollicular fibrosis. Therefore, Patel et al. concluded that the process underlying this event might be follicular with varying degrees of lichenoid inflammation, resembling the rare Graham–Little–Piccardi–Lassueur syndrome.

As concerns ACRs in general, cross-intolerance between imatinib and second-generation TKIs was regarded as rare [23]. Interestingly, among the nine patients with KP-like lesions examined by Patel et al., six of them had previously been treated with imatinib without experiencing similar reactions [24]. The absence of cross-intolerance between imatinib and other TKIs was confirmed by other reports [36,37,39,40,41]. In such cases, imatinib had been stopped because of resistance or intolerance [36,37,39,40,41] (Table 1). Among the patients examined by Patel et al. [24], two patients had KP-like lesions with both dasatinib and nilotinib, whereas in another patient the rash resolved after switching from nilotinib to dasatinib. In another child, there was a history of mild pruritic rashes that developed a few weeks after starting dasatinib and became worse, leading to the development of KP-like lesions after the switch to nilotinib [39].

In the publications cited in Table 1, information about medical history was not constantly outlined. However, the reported data indicated no significant personal and/or familial history of KP and/or other skin disorders [24,35,36,37,38,40,41]. The onset of the reaction varied from a few days to 6 months. Some patients had asymptomatic lesions without relevant inflammatory signs or hair thinning [35,36,41,42], as in our case, with similar histopathological findings that were available for two patients [35,41]. In fact, in both reports [35,41], histopathological features were suggestive of KP, being characterized by keratotic plugs within the follicle and mild lymphocytic infiltration without fibrosis.

In the literature review, some publications underlined the presence of pruritus [22,24,39,40] and case reports documented the association with hair loss and perifollicular fibrosis (KP atrophicans) [37], non-scarring hair thinning [24,38,40], alopecia areata [38] cicatricial alopecia resembling frontal fibrosing alopecia [40], oral erosive lichen planus, and facial acneiform lesions [24]. The association with follicular lichenoid lesions was pointed out by Patel et al. [24], as previously mentioned. Follicular atrophy and/or fibrotic changes were sometimes reported [23,24,37,39,40], and these histopathological findings were sometimes seen in patients with hair loss [24,37,40].

Data on the management and outcome of the cutaneous reaction were present only in a few articles. In general, there was a tendency toward the use of emollients or keratolytic agents for lesions with no or minimal symptoms and without relevant signs of inflammation, whereas topical steroids and other drugs were used for manifestations characterized by pruritus and inflammatory changes. Heterogeneous results were obtained. Improvement with dose reduction [24] and resolution with drug discontinuation were observed [24,39,41]. Rechallenge with the culprit drug was found to induce the reappearance of cutaneous lesions [39,41], as in our case.

Some reports stressed the importance of continuing the administration of TKIs, as the cutaneous manifestations were often considered mild and tolerable, whereas the role of TKIs was fundamental [35,36,38,41,42].

The pathogenesis of KP-like rashes induced by TKIs is still unknown and was attributed to their mechanism of action. Most articles focused on nilotinib and dasatinib due to the higher rates of reporting for such agents. Nilotinib and dasatinib are potent inhibitors of BCR-ABL but their activity is not simply directed at tumor cells, as they also act against other targets, such as additional kinases. The majority of such kinases are active on the skin, and their inhibition could have a role in epidermal homeostasis and in the development of the rash [22,26]. Off-target pathways may be affected by TKIs, leading to downstream effects responsible for various adverse events.

A diffuse folliculocentric keratotic reaction mimicking KP is commonly observed in patients treated with RAF inhibitors [26]. KP and other cutaneous reactions caused by RAF inhibitors resemble skin changes included in the clinical spectrum of the so-called RASopahies, rare genetic disorders with activating *RAS/MAPK* germline mutations, and RAF inhibitors can cause paradoxical RAS activation [43,44]. It was shown that imatinib, nilotinib, and dasatinib possess weak off-target activity against RAF and can drive paradoxical RAF/MEK/ERK pathway activation [24,45]. Another mechanism hypothesized for vemurafenib-induced KP was the activation of downstream protein kinase R-like endoplasmic reticulum kinase in follicular epithelial cells [34]. A case of new-onset KP has been described in a patient after discontinuation of erlotinib, a TKI that inhibits EGFR, and this event might result from the interference with EGFR in the skin and hair follicle signaling [46]. Sorafenib, a multi-kinase inhibitor targeting C-RAF, BRAF, VEGFRs, PDGFR-beta, RET, KIT, and FLT-3 but not BCR-ABL, can cause generalized KP with histological features typical for KP [47].

Our literature search in the PubMed database did not disclose reports of KP or KP-like eruptions related to treatment with imatinib for CML. We cannot rule out that KP-like eruptions during treatment with imatinib can be milder than those induced by newer-generation BCR-ABL inhibitors, and for this reason they can be neglected and underestimated in clinical practice. Nevertheless, the higher frequency of this eruption in patients receiving newer generation ATP-competitive TKIs, especially nilotinib and dasatinib, as compared with imatinib-treated patients, might be related to the different mechanisms of action and target spectrum of these drugs. Different activities on molecular targets other than BCR-ABL, including kinase and non-kinase targets, can also justify differences in the rate of development of KP-like eruptions among BCR-ABL TKIs.

## Figures and Tables

**Figure 1 jcm-13-00032-f001:**
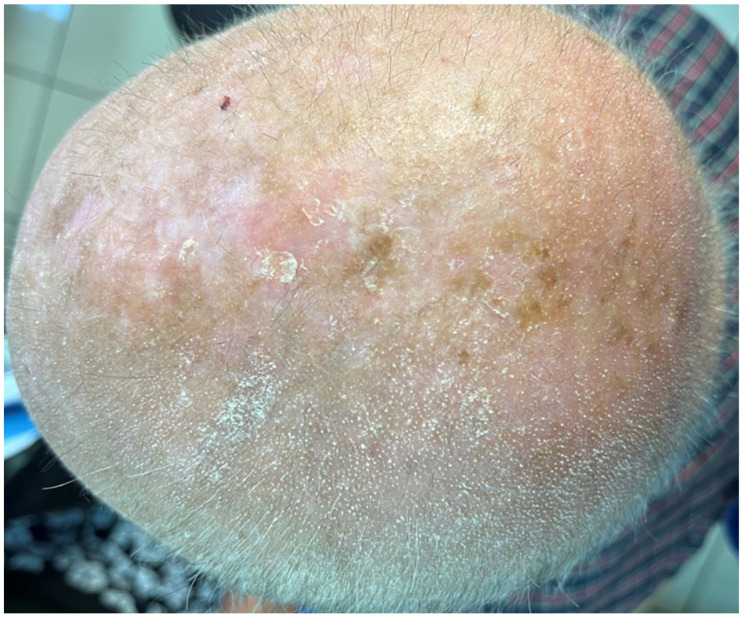
Multiple tiny follicular, white-colored lesions affecting the patient’s scalp.

**Figure 2 jcm-13-00032-f002:**
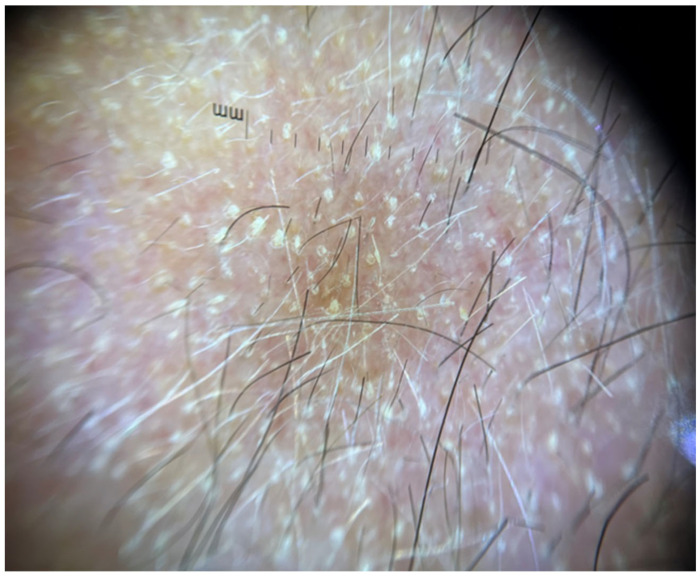
Dermoscopic findings consistent with follicular hyperkeratosis in an affected area of the scalp.

**Figure 3 jcm-13-00032-f003:**
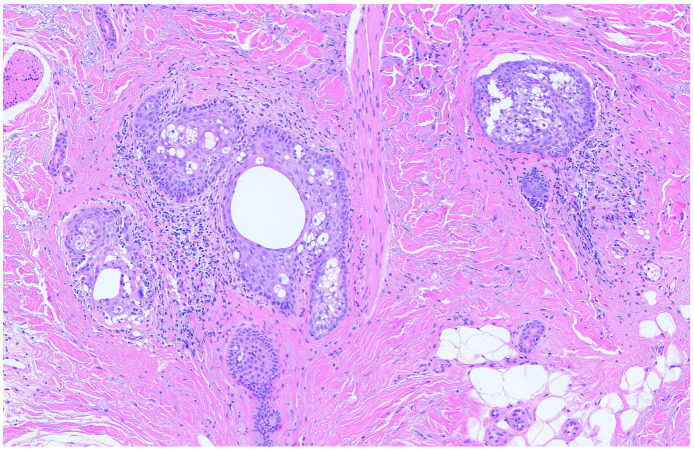
Histological photomicrograph showing features suggestive of keratosis pilaris within a dermis with diffuse and severe sclero-elastotic damage (Hematoxylin-Eosin, Original Magnification 10×).

**Figure 4 jcm-13-00032-f004:**
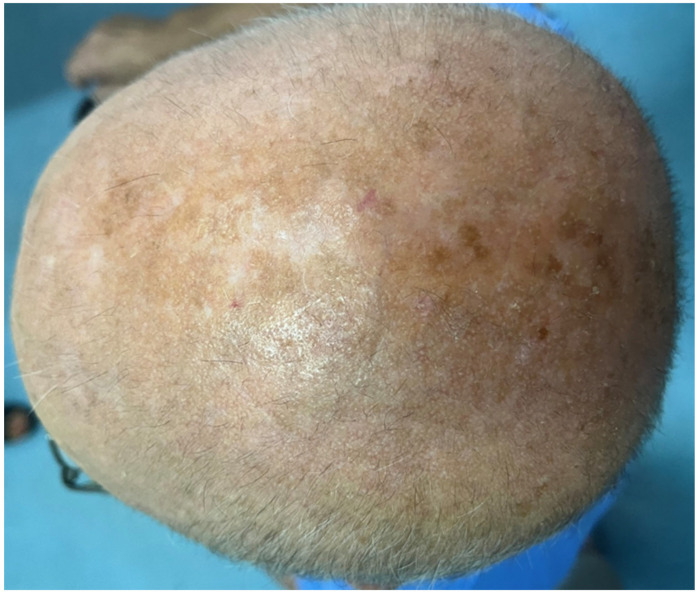
Clinical picture of the patient’s scalp after discontinuation of treatment with imatinib.

**Figure 5 jcm-13-00032-f005:**
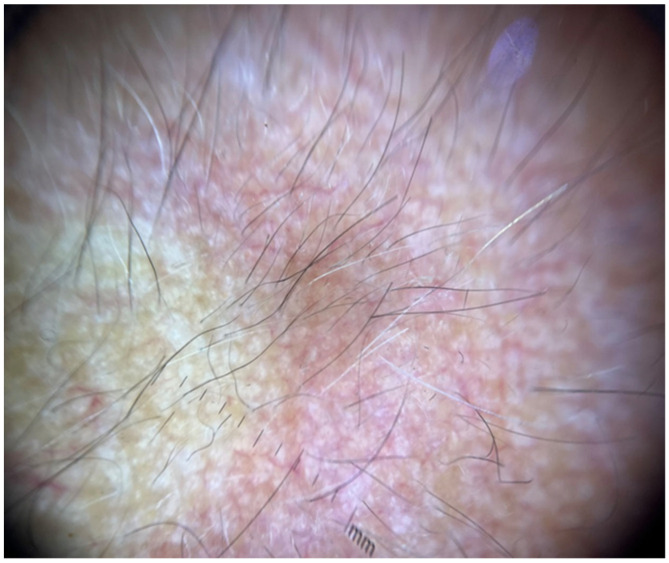
Dermoscopic findings after discontinuation of imatinib in the same area shown in Figure 2.

**Table 1 jcm-13-00032-t001:** Reported cases of KP or KP-like eruption induced by BCR-ABL TKIs in patients with CML.

First Author [Reference]	N.	Sex	Age (Years)	Previous TKI	Culprit TKI (Daily Dose at Rash Onset)	KP/KP-like Eruption			Other Findings	Histopathology
						Clinical Details	Localization	Time to Onset (after TKI Start)		
Drucker [22]	2	F	52	-	N (500 mg)	CTCAE grade 2	Face, extremities	NR	Erythematous papules Tenderness, pain, pruritus	NA
		M	69	-	D (100 mg)	CTCAE grade 2	Trunk, extremities	NR	Pustules Pruritus	Milium
Delgado [23]	9	NR	NR	-	N or D (not specified)	Extensive KP	-	NR		Follicular atrophy, perifollicular fibrosis (biopsy in 4 patients)
Shimizu [35]	1	F	68	-	N (600 mg)	Asymptomatic, milium-sized, skin-colored keratotic papules	Trunk, extremities	6 months		Keratotic plug within the follicle, slight perifollicular lymphocytic infiltration
Patel [24]	9	F	56	I *	N (400 mg b.i.d.)	KP-like eruption (from subtle follicular prominence to flat-topped follicular erythematous to violaceous papules with mild scale)	Face, trunk, and/or extremities (usually forehead, temples, lateral cheeks, proximal extremities; occasionally, trunk)	1 month	Pruritus Scalp LPP-like eruption with nonscarring alopecia Follicular erythematous papules of eyebrows with nonscarring alopecia	Arm: KP-like or suppurative folliculitis Scalp: subtle LPP-like
		F	54	I *	N (400 mg b.i.d.)			2 months	Pruritus Scalp LPP-like eruption LP-like lesions on trunk Oral erosive LP	Arm: KP-like Scalp: subtle LPP-like Trunk: LP
		F	55	I *, D	P (30 mg; initially 4 mg)			weeks after dose escalation	Scalp LPP-like eruption Follicular erythematous papules of eyebrows with nonscarring alopecia	NA
		M	53	I *, D, N	P (45 mg)			2 months	Pruritus Scalp LPP-like eruption	NA
		F	41	-	D (100 mg)			2–3 months	Pruritus Scalp LPP-like eruption Facial acneiform eruption	Thigh: KP-like with fibrosis
		F	51	-	N (300 mg b.i.d.)			2 months	Pruritus	NA
		M	49	-	N (300 mg b.i.d.)			1.5 months	Pruritus	NA
		F	56	I ^	N (400 mg)			days after restart	Pruritus Scalp LPP-like eruption with scarring alopecia Follicular erythematous papules of eyebrows with nonscarring alopecia	
					D (100 mg)			continued		Back: KP-like with vacuolar changes Scalp: LPP-like
		M	61	I *	D (180 mg; initially 140 mg)			weeks after dose escalation	Pruritus	
					N (400 mg b.i.d.)			continued	Pruritus Scalp LPP-like eruption with scarring alopecia Follicular erythematous papules of eyebrows with nonscarring alopecia	Trunk: perifolliculitis Scalp: LPP-like
Leong [36]	1	M	27	I *	N (400 mg b.i.d.)	Non-pruritic, rough, brown, follicular papules	Trunk, limbs (especially the extensor portion of upper limbs)	3 days	No alopecia	NA
Khetarpal [37]	1	M	46	I *	N (400 mg b.i.d.)	Prominent perifollicular pink papules	Arms, chest, back, shoulders, legs	2 months	Complete or partial hair loss in affected areas; lateral thinning of eyebrows; follicular accentuation on the forehead; skin dryness and roughness	Prominent perifollicular fibrosis extending to the dermis (KP atrophicans)
Tawil [38]	1	M	45	-	N (300 mg b.i.d.)	Asymptomatic, keratotic, red-brown follicular papules	Limbs (especially the extensor surfaces of upper limbs)	4 months	Nonscarring eyebrow hair loss Thinning of chest hair Autoresolutive alopecia areata of the wrist	NA
Oro-Ayude [39]	1	M	14	I *, D	N (NR)	1–2 mm, rough, skin-colored, folliculocentric papules on an erythematous base	Generalized, with prominent involvement of the eyebrows, ears, extensor surfaces of upper limbs	several days	Pruritus	Dilated follicular infundibulum, basket-weave orthokeratosis, follicular plug, perifollicular concentric fibrosis, mild perivascular lymphocytic inflammation
Frioui [40]	1	F	49	I *	N (NR)	Small, rough, skin-colored keratotic follicular papules	Scalp, face, neck, trunk, extremities	4 months	Pruritus Nonscarring hair thinning in eyebrows, axillae, and pubic area Cicatricial alopecia resembling frontal fibrosing alopecia	Plugging of individual hair follicles. Scalp: largely unaffected epidermis, rarefaction of hair follicles with isthmic atrophy, keratotic plugs within the follicle, perivascular lymphocytic infiltrate
Kowe [41]	1	F	40	N ^§^, I **	N (300 mg b.i.d.)	Asymptomatic, small raised, keratotic follicular papules	Face, extensor aspects of upper extremities, upper back	10 days (after restarting)		Follicular hyperkeratosis, dilated follicular infundibulum with keratotic plugging, mild perivascular lymphocytic infiltrate in upper dermis
Jimenez-Cauhe [42]	1	M	17	-	N (350 mg b.i.d.)	Non-pruritic, millimetric, skin-colored, follicular-centered papules with rough surface	Lateral aspects of face, upper limbs, trunk, proximal portion of lower limbs	6 days		NA

b.i.d., twice daily; CML, chronic myeloid leukemia; CTCAE, National Cancer Institute’s Common Terminology Criteria for Adverse Events; D, dasatinib; F, female; I, imatinib; KP, keratosis pilaris; LP, lichen planus; LPP, lichen planopilaris; M, male; N, nilotinib; NA, not available (biopsy non performed or refused by the patient); P, ponatinib; NR, not reported; TKI, tyrosine kinase inhibitor. Discontinuation of imatinib because of * resistance; ^ nondermatological intolerance; ** development of drug-induced rash with eosinophilia and systemic symptoms syndrome; ^§^ Nilotinib initially stopped after 3 months because of localized KP, which resolved with drug holiday.

## Data Availability

Data are contained within the article.

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
