# Peer review of "Keratosis Pilaris-like Eruption during Treatment of Chronic Myeloid Leukemia with Tyrosine Kinase Inhibitors: Literature Review and Report of a Case Related to Imatinib"

_jcm, 2023, doi:10.3390/jcm13010032_

Round 1
Reviewer 1 Report
Comments and Suggestions for Authors
Ambrogio and colleagues present here a single case report of keratosis pilaris like eruption in a patient with CML. The case is well presented, the literature review is decent and the pictures add positively to the manuscript.
Adverse cutaneous reactions like KP are not uncommon with TKIs used in CML and have been extensively published before. It is understandable that therapy with imatinib is often the most tolerated and KP like eruptions have not/rarely been reported with imatinib. However frontline therapy with imatinib has now given way to second generation TKIs in lieu of faster and deeper response, though imatinib continues to be a very important drug in against CML . The present case of KP with imatinib is important but not novel, however this will add to the clinicians knowledge who can encounter such situations when treating a patient with CML.
Major comments:
1) Please consider reducing the elaborate section on TKIs in CML in the introduction. The crux of the paper is about ACR to imatinib with a focus on KP and the detailed introduction about TKIs in CML, including that of asciminib, is possibly unnecessary.
2) Please consider citing and discussing recent reviews that discuss about drug toxicity in CML and TKI dose reduction from the aspect of drug toxicity. One important such review is mentioned below:
"Management of Chronic Myeloid Leukemia in 2023: Common ground and common sense" (PMID: 37088793)
Minor Comments:
1) A small flowchart depicting the skin toxicity with imatinib and its resolution with imatinib drug holiday could be considered in the case presentation.That might help in easier understanding for the readers.
Author Response
Reviewer 1
Comments and Suggestions for Authors
Ambrogio and colleagues present here a single case report of keratosis pilaris like eruption in a patient with CML. The case is well presented, the literature review is decent and the pictures add positively to the manuscript.
Adverse cutaneous reactions like KP are not uncommon with TKIs used in CML and have been extensively published before. It is understandable that therapy with imatinib is often the most tolerated and KP like eruptions have not/rarely been reported with imatinib. However frontline therapy with imatinib has now given way to second generation TKIs in lieu of faster and deeper response, though imatinib continues to be a very important drug in against CML.The present case of KP with imatinib is important but not novel, however this will add to the clinicians knowledge who can encounter such situations when treating a patient with CML.
Re: Thank you
Major comments:
1) Please consider reducing the elaborate section on TKIs in CML in the introduction. The crux of the paper is about ACR to imatinib with a focus on KP and the detailed introduction about TKIs in CML, including that of asciminib, is possibly unnecessary.
Re: We eliminated some sentences in the introduction, as suggested, in particular those related to the current indication of TKIs as frontline or second-line therapy in CML and other sentences regarding asciminib.
2) Please consider citing and discussing recent reviews that discuss about drug toxicity in CML and TKI dose reduction from the aspect of drug toxicity. One important such review is mentioned below:
"Management of Chronic Myeloid Leukemia in 2023: Common ground and common sense" (PMID: 37088793)
Re: Thank you very much for your suggestion. We briefly mentioned the possibility of managing toxicities with dose reduction, citing the interesting review that was indicated.
Because of the above-mentioned changes, we modified references, eliminating the previous references n. 6 and n. 28 and substituting them with new ones.
Minor Comments:
- A small flowchart depicting the skin toxicity with imatinib and its resolution with imatinib drug holiday could be considered in the case presentation.That might help in easier understanding for the readers.
Re: Thank you. However, we tried to make a flowchart but it appeared to be redundant. We tried to explain in detail the different phases of the cutaneous drug reaction in our patient in the first part of the discussion.
Reviewer 2 Report
Comments and Suggestions for Authors
They suggested Keratosis pilaris-like eruption is a skin condition characterized by small, rough bumps on the skin, typically on the arms, thighs, and buttocks. It has been observed in patients undergoing treatment with tyrosine kinase inhibitors (TKIs) for chronic myeloid leukemia (CML) and is one of the side effects of treatment, it appears to be more common in patients treated with newer generation ATP-competitive TKIs, such as nilotinib and dasatinib, compared to imatinib-treated patients. Dermatologists can manage keratosis pilaris-like eruption in patients undergoing tyrosine kinase inhibitor (TKI) therapy by providing symptomatic relief and addressing the underlying cause. Topical treatments such as emollients, keratolytics, and topical corticosteroids can help to reduce the appearance of the bumps and relieve itching. Oral antihistamines may also be prescribed to alleviate itching. In severe cases, systemic corticosteroids or other immunosuppressive agents may be necessary. It is important to monitor patients for signs of infection or other complications, as TKIs can affect the immune system.
1. Authors explained direct correlation between the imatinib drug and the skin reaction based on key factors like
2. Temporal Association: The onset of the skin reaction occurring after the initiation or reintroduction of imatinib and the subsequent remission upon discontinuation or dose reduction strongly suggests a drug-related cause.
3. Exclusion of Other Causes: Clinically and histologically ruling out other potential dermatologic conditions that could mimic or cause similar symptoms is essential. In this case, the differential diagnosis was made based on the absence of specific histopathological findings seen in other skin conditions.
4. Rechallenge Test: Reintroducing the drug and observing a milder recurrence of the skin manifestations further supports the association. While this is not always feasible or recommended in clinical practice due to potential risks, it can sometimes strengthen the evidence linking the drug to the reaction.
On summarizing, upon discontinuation of imatinib, there was complete clinical remission, indicating a direct correlation between the drug and the skin reaction. However, upon reintroduction of imatinib at a reduced dosage, the skin manifestations reappeared, albeit in a milder form, and were manageable
Comments on the Quality of English LanguageThey suggested Keratosis pilaris-like eruption is a skin condition characterized by small, rough bumps on the skin, typically on the arms, thighs, and buttocks. It has been observed in patients undergoing treatment with tyrosine kinase inhibitors (TKIs) for chronic myeloid leukemia (CML) and is one of the side effects of treatment, it appears to be more common in patients treated with newer generation ATP-competitive TKIs, such as nilotinib and dasatinib, compared to imatinib-treated patients. Dermatologists can manage keratosis pilaris-like eruption in patients undergoing tyrosine kinase inhibitor (TKI) therapy by providing symptomatic relief and addressing the underlying cause. Topical treatments such as emollients, keratolytics, and topical corticosteroids can help to reduce the appearance of the bumps and relieve itching. Oral antihistamines may also be prescribed to alleviate itching. In severe cases, systemic corticosteroids or other immunosuppressive agents may be necessary. It is important to monitor patients for signs of infection or other complications, as TKIs can affect the immune system.
1. Authors explained direct correlation between the imatinib drug and the skin reaction based on key factors like
2. Temporal Association: The onset of the skin reaction occurring after the initiation or reintroduction of imatinib and the subsequent remission upon discontinuation or dose reduction strongly suggests a drug-related cause.
3. Exclusion of Other Causes: Clinically and histologically ruling out other potential dermatologic conditions that could mimic or cause similar symptoms is essential. In this case, the differential diagnosis was made based on the absence of specific histopathological findings seen in other skin conditions.
4. Rechallenge Test: Reintroducing the drug and observing a milder recurrence of the skin manifestations further supports the association. While this is not always feasible or recommended in clinical practice due to potential risks, it can sometimes strengthen the evidence linking the drug to the reaction.
On summarizing, upon discontinuation of imatinib, there was complete clinical remission, indicating a direct correlation between the drug and the skin reaction. However, upon reintroduction of imatinib at a reduced dosage, the skin manifestations reappeared, albeit in a milder form, and were manageable.
Author Response
Reviewer 2
Comments and Suggestions for Authors
They suggested Keratosis pilaris-like eruption is a skin condition characterized by small, rough bumps on the skin, typically on the arms, thighs, and buttocks. It has been observed in patients undergoing treatment with tyrosine kinase inhibitors (TKIs) for chronic myeloid leukemia (CML) and is one of the side effects of treatment, it appears to be more common in patients treated with newer generation ATP-competitive TKIs, such as nilotinib and dasatinib, compared to imatinib-treated patients. Dermatologists can manage keratosis pilaris-like eruption in patients undergoing tyrosine kinase inhibitor (TKI) therapy by providing symptomatic relief and addressing the underlying cause. Topical treatments such as emollients, keratolytics, and topical corticosteroids can help to reduce the appearance of the bumps and relieve itching. Oral antihistamines may also be prescribed to alleviate itching. In severe cases, systemic corticosteroids or other immunosuppressive agents may be necessary. It is important to monitor patients for signs of infection or other complications, as TKIs can affect the immune system.
- Authors explained direct correlation between the imatinib drug and the skin reaction based on key factors like
- Temporal Association:The onset of the skin reaction occurring after the initiation or reintroduction of imatinib and the subsequent remission upon discontinuation or dose reduction strongly suggests a drug-related cause.
- Exclusion of Other Causes:Clinically and histologically ruling out other potential dermatologic conditions that could mimic or cause similar symptoms is essential. In this case, the differential diagnosis was made based on the absence of specific histopathological findings seen in other skin conditions.
- Rechallenge Test:Reintroducing the drug and observing a milder recurrence of the skin manifestations further supports the association. While this is not always feasible or recommended in clinical practice due to potential risks, it can sometimes strengthen the evidence linking the drug to the reaction.
On summarizing, upon discontinuation of imatinib, there was complete clinical remission, indicating a direct correlation between the drug and the skin reaction. However, upon reintroduction of imatinib at a reduced dosage, the skin manifestations reappeared, albeit in a milder form, and were manageable
Re: Thank you
As moderate editing of English language was recommended, we made some changes in the text.